# Fast variable selection makes Karhunen-Loève decomposed Gaussian process BSS-ANOVA a speedy and accurate choice for dynamic systems identification

## Abstract

Many approaches for scalable GPs have focused on using a subset of data as inducing points. Another promising approach is the Karhunen-Loève (KL) decomposition, in which the GP kernel is represented by a set of basis functions which are the eigenfunctions of the kernel operator. Such kernels have the potential to be very fast, and do not depend on the selection of a reduced set of inducing points. However KL decompositions lead to high dimensionality, and variable selection thus becomes paramount. This paper reports a new method of forward variable selection, enabled by the ordered nature of the basis functions in the KL expansion of the Bayesian Smoothing Spline ANOVA kernel (BSS-ANOVA), coupled with fast Gibbs sampling in a fully Bayesian approach. It quickly and effectively limits the number of terms, yielding a method with competitive accuracies, training and inference times for tabular datasets of low feature set dimensionality. The new algorithm determines how high the orders of included terms should reach, balancing model fidelity with model complexity using $L^0$ penalties inherent in Bayesian and Akaike information criteria. The inference speed and accuracy makes the method especially useful for modeling dynamic systems, by modeling the derivative in a dynamic system as a static problem, then integrating the learned dynamics using a high-order scheme. The methods are demonstrated on two dynamic datasets: a 'Susceptible, Infected, Recovered' (SIR) toy problem, with the transmissibility used as forcing function, along with the experimental 'Cascaded Tanks' benchmark dataset. Comparisons on the static prediction of derivatives are made with a random forest (RF), a residual neural network (ResNet), and the Orthogonal Additive Kernel (OAK) inducing points scalable GP, while for the timeseries prediction comparisons are made with LSTM and GRU recurrent neural networks (RNNs). The GP outperforms the RF and ResNet on the static estimation, and is comparable to OAK. In dynamic systems modeling it outperforms both RNNs, while performing many orders of magnitude fewer calculations. For the SIR test, which involved prediction for a set of forcing functions qualitatively different from those appearing in the training set, BSS-ANOVA captured the correct dynamics while the neural networks failed to do so.

## 1 Karhunen-Loève decomposed Gaussian processes

### 1.1 Gaussian process fundamentals

Gaussian processes (GPs) are stochastic functions that are engines for nonparametric regression. Initially developed for modeling and interpolation in geographic information systems datasets,

applications have multiplied across many fields of data science. A key advantage of the GP is its broad, continuous nonparametric support and the frequent amenability of different GP kernels to precise analysis.

A GP is Gaussian in that it is a covariance model linking pairs of points on functional draws. As such a GP is completely described by a mean function (often zero in the prior) and covariance kernel. The most famous and perhaps simplest of the covariance kernels is the squared exponential:

$$\kappa(x, x') = \varsigma^2 \exp\left[\frac{(x - x')^2}{\xi}\right] \tag{1}$$

where the sill $\varsigma^2$ and range $\xi$ parameters determine the scale and smoothness of the draws. In a typical implementation modeling a static dataset $Z$, the statistical model

$$Z = \delta(\mathbf{x}|\varsigma^2, \xi) + \epsilon \tag{2}$$

with $\epsilon$ an observation error process, is first used to infer the hyperparameters, after which predictions conditioned on the training dataset can be made. The draws on the squared exponential GP – a limiting case of the Matérn covariance family – are infinitely differentiable.

From a practical standpoint the training of the above GP is $\mathcal{O}(N^3)$, requiring a Cholesky decomposition of the full covariance matrix. This limits the use of the GP to moderately-sized datasets, generally of a thousand instances or fewer.

## 1.2 Scalable Gaussian processes with inducing points

Liu et al. [2020] provide a thorough overview of efforts that aim to improve scalability while maintaining prediction accuracy using global kernel approximations derived in some sense from a set of $M << N$ inducing points [Chalupka et al., 2013, Quinonero-Candela and Rasmussen, 2005, Deisenroth and Ng, 2015, Rasmussen and Ghahramani, 2001, Wang et al., 2022]. Generally the goal is to approximate the full-rank kernel matrix with local approximations. Of particular note is a $\mathcal{O}(N)$ method that directly estimates the covariance with training and inference times that limits the increase in $M$ for large $N$ developed by Wilson et al. [2015]. Some methods employ ANOVA decompositions to the full kernel which break out contributions in terms of features and their combinations:

$$\kappa(\mathbf{x}, \mathbf{x}') = \sum_{i=1}^{n} \kappa_i(x_i, x_i') + \sum_{i=1}^{n-1} \sum_{j=i+1}^{n} \kappa_i(x_i, x_i')\kappa_j(x_j, x_j') + \cdots \tag{3}$$

which presents opportunities for variable selection [Duvenaud et al., 2011]; of particular note is the recent Orthogonal Additive Kernel (OAK) which orthogonalizes the kernels in (3) in order to minimize overlap between main effects and higher-order interactions [Lu et al., 2022].

## 1.3 Karhunen-Loève decomposition and BSS-ANOVA

Another approach to scalability in GPs that is distictive to the inducing points approach is the Karhunen-Loève (KL) expansion, in which the kernel is expressed in terms of a sum over its eigenfunctions:

$$\delta(x; \boldsymbol{\beta}) \sim MVN(0, \kappa) = \sum_i \beta_i \phi_i(x) \tag{4}$$

where

$$\phi_i(x) = \sqrt{\lambda_i} u_i(x) \tag{5}$$

$$\int \kappa(x, x') u_i(x') dx' = \lambda_i u(x) \tag{6}$$

$$\beta_i \sim N(0, \lambda_i) \tag{7}$$

Such methods have the potential to be fast: $\mathcal{O}(NP)$ in training and $P$ per point for inference, where $P$ is the number of terms in the expansion. However such kernels have not been the subject of much research in machine learning contexts generally. The main issues are tractable calculation of the basis functions $\{\phi_i\}$ and dimensionality issues [Greengard and O'Neil, 2021].

In 2009 Reich et al. [2009] introduced the Bayesian Smoothing Spline ANOVA (BSS-ANOVA) kernel, which is subject first to an ANOVA decomposition, followed by a KL decomposition. The core of the BSS-ANOVA kernel is:

$$\kappa_1(x, x') = \mathcal{B}_1(x)\mathcal{B}_1(x') + \mathcal{B}_2(x)\mathcal{B}_2(x') + \frac{1}{24}\mathcal{B}_4(|x - x'|) \tag{8}$$

where $\mathcal{B}_k$ is the $k^{\text{th}}$ Bernoulli polynomial, defined by the generating function

$$\frac{te^{tx}}{e^t - 1} = \sum_{i=0}^{\infty} \mathcal{B}_i(x)\frac{t^i}{i!} \tag{9}$$

yielding

$$\mathcal{B}_1(x) = x - \frac{1}{2} \tag{10}$$

$$\mathcal{B}_2(x) = x^2 - x + \frac{1}{6} \tag{11}$$

$$\mathcal{B}_4(x) = x^4 - 2x^3 + x^2 - \frac{1}{30} \tag{12}$$

This kernel is effectively a sum of a non-stationary quadratic response surface – corresponding to the first two terms in (8) – and a stationary deviation (the final term). As in (3), covariances for higher-order interactions are constructed with dyadic products of the main effect covariance:

$$\kappa_2([x_j, x_k], [x'_j, x'_k]) = \kappa_1(x_j, x'_j)\kappa_1(x_k, x'_k) \tag{13}$$

and so on for higher-order interactions. Terms are then multiplied by scaling hyperparameters and added together to produce the full kernel:

$$\kappa = \sigma_0^2\tau_0^2 + \sigma_1^2\tau_1^2 \sum_{i=1}^{n} \kappa_{1,i} + \sigma_2^2\tau_2^2 \sum_{i=1}^{n-1} \sum_{j=i+1}^{n} \kappa_{2,ij} + \cdots \tag{14}$$

The kernel so constructed is supported by a second-order Sobolev space [Reich et al., 2009], which is a very broad and dense set of continuous functions.

Building the kernel in this fashion effectively addresses the problem of generating the eigenfunctions from the KL decomposition: because all of the terms in (14) are based on the generative kernel (8), The KL decomposition of 14 will depend only on eigenfunctions of $\kappa_1$. Additionally if all input features are normalized to an $[0, 1]$ interval (we restrict the discussion to continuous input features for now), then it is only necessary to compute a single set of basis functions $\{\phi_i\}$. The decomposed BSS-ANOVA GP is written:

$$\delta(\mathbf{x}; \boldsymbol{\beta}) = \beta_0 + \sum_{i=1}^{n} \sum_{k=1}^{\infty} \beta_{ik}\phi_k(x_i) + \sum_{i=1}^{n-1} \sum_{j=i+1}^{n} \sum_{k=1}^{\infty} \sum_{l=1}^{\infty} \beta_{ik,jl}\phi_k(x_i)\phi_l(x_j) + \cdots \tag{15}$$

Given the assumption

$$\sigma_0^2\tau_0^2 = \sigma_1^2\tau_1^2 = \sigma_2^2\tau_2^2 = \cdots = \sigma^2\tau^2 \tag{16}$$

then the priors for the coefficients $\boldsymbol{\beta}$ are iid normal

$$\beta_{\cdot k} \sim N(0, \sigma^2\tau^2) \tag{17}$$

Following [Reich et al., 2009] we generate the set $\{\phi_i\}$ by producing $\kappa_1$ for a dense grid consisting of 500 intervals on $[0, 1]$, eigendecompose and fit to cubic splines. Figure 1 shows the first 6 basis functions. These basis functions are nonparametric, pairwise orthogonal, and ordered: note the increase in frequency and decrease in amplitude as the orders increase.

## 2  Variable selection

It's clear from (15) that the number of terms in the expansion can increase rapidly, even for low-dimensional input spaces. A key component of applying the GP to a modeling problem is thus the selection of terms. Effectively we seek to minimize the objective function

$$\Phi(\boldsymbol{\beta}) = ||Z - \delta(\mathbf{x}; \boldsymbol{\beta})||^2 + \zeta(\boldsymbol{\beta}) \tag{18}$$

where $\zeta$ is a penalty function which leads to a sufficiently sparse solution.

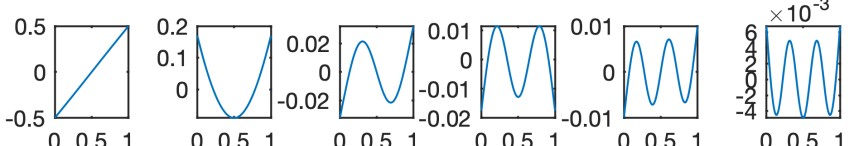

Figure 1: The first six basis functions of the KL-decomposed BSS-ANOVA kernel. The basis is nonparametric, spectral, pairwise orthogonal and ordered.

## 2.1 Indicator variable methods

Reich et al. [2009] took a hierarchical Bayesian approach to the problem, estimating a separate variance $\tau^2$ for each term in the expansion, which is in turn expressed in terms of an indicator variable with a Bernoulli prior. This approach, like other 'indicator variable' methods, accomplishes the variable selection and the training simultaneously and comprehensively, at the cost of requiring a large number of variables in the prior model and a computationally onerous Markov chain Monte Carlo (MCMC) sampling procedure.

Other sparse optimization methods such as ridge regression or LASSO share the limitation that many high-order terms must be included in the initial model before downselection occurs.

## 2.2 Forward variable selection

The ordered and orthogonal nature of the basis functions suggests a forward variable selection approach. Rewriting the model (15) for a basis function set of maximum order $q$,

$$\delta(\mathbf{x}; \boldsymbol{\beta}) = \beta_0 + \sum_{i=1}^{n} \sum_{k=1}^{q} \beta_{ik} \phi_k(x_i) + \sum_{i=1}^{n-1} \sum_{j=i+1}^{n} \sum_{k=1}^{q} \sum_{l=1}^{q} \beta_{ik,jl} \phi_k(x_i) \phi_l(x_j) + \cdots \quad (19)$$

then considering a model building procedure which increases $q$ stepwise starting with $q = 1$ reveals that each subsequent step adds $n$ main effect terms (each depending on a single input), $\binom{n}{2}[2(q-1)+1]$ two-way interactions, and $\binom{n}{3}[3(q-1)^2 + 3(q-1) + 1]$ three-way interactions. As the model order increases the $L^2$ truncation error for the full kernel decreases as (for the case of a single input)[Greengard and O'Neil, 2021]:

$$||\kappa(x, x') - \sum_{i=1}^{q} \phi_i(x)\phi_i(x')|| < \Big( \sum_{i=q+1}^{\infty} \lambda_i^2 \Big)^{1/2} \quad (20)$$

Since the eigenvalues of the BSS-ANOVA kernel decomposition decrease quickly with increasing order, an approach to the optimization problem (18) focusing on low-order models will sacrifice little in the way of accuracy while realizing significant advantages in computing time.

The design and implementation of such an approach is the main contribution of this work. It approaches the optimization of (18) with an iterative process, finding the most efficient truncation of the system while evaluating the cost function only for candidate models with *fewer* terms than the optimum truncation. The method is fully Bayesian, with a fast Gibbs sampling procedure at its core. As such the form of the cost function is also Bayesian in nature, taking the form of the Bayesian or Akaike information criteria (BIC/AIC), which incorporate $L^0$ penalties.

### 2.2.1 Gibbs sampling

Given a statistical model

$$z_i = \delta_i(\mathbf{x}_i; \boldsymbol{\beta}) + \epsilon \quad (21)$$

with $\epsilon$ a white noise observation error, and a given truncation to the KL expansion (15), the model is linear in the coefficients $\beta$ and Gibbs sampling can be used to estimate parameters in a fully Bayesian methodology.

If the variance of the observation error is $\sigma^2$, with inverse gamma prior $\sigma^2 \sim IG(a, b)$, with $a$ and $b$ the shape and scale parameters, respectively; and if $\tau^2$ has inverse gamma prior $\tau^2 \sim IG(a_\tau, b_\tau)$,

then an iterative Gibbs sampler can be devised such that for fixed $\{\sigma^2, \tau^2\}$, $\beta \sim MVN(\mu, \Sigma)$, with

$$\mu = (X^T X + 1/\tau^2 I)^{-1} X^T Z \tag{22}$$

$$\Sigma = \sigma^2 \Big( X^T X + 1/\tau^2 I \Big)^{-1} \tag{23}$$

where $X \in \mathbb{R}^{N \times P}$ is a matrix constructed from the basis functions, whose rows correspond to instances and columns to terms in the expansion. For fixed $\{\beta, \tau^2\}$, $\sigma^2 \sim IG(a^*, b^*)$, with

$$a^* = a + 1 + N/2 + P/2 \tag{24}$$

$$b^* = b + \frac{1}{2} \Big[ (\mu - \beta)^T (X^T X + 1/\tau^2 I)(\mu - \beta) + Z^T Z - \mu^T X^T Z \Big] \tag{25}$$

For fixed $\{\beta, \sigma^2\}$, $\tau^2 \sim IG(a_\tau^*, b_\tau^*)$, with

$$a_\tau^* = a_\tau + (P - 1)/2 \tag{26}$$

$$b_\tau^* = b_\tau + \frac{1}{2\sigma^2} \beta^T \beta \tag{27}$$

This algorithm is implemented in the routine 'gibbs_Xin' in the supplement.

### 2.2.2 Optimization algorithm

The algorithm constructs models with terms having up to three-way interactions. Terms are added in stages labeled by an integer "index" that initializes at 1. At each stage, a series of substages cycle through all permutations of basis function orders that sum up to that stage's index. Stage 1 adds only first order main effects. Stage 2 adds second order main effects and first order two way interactions – corresponding to $\phi_1(x_i)\phi_1(x_j)$ – in two separate substages. The substages always occur such that terms involving lower-order basis functions (for example in the case of stage 2, this is the first order two-way interactions) come first. Each substage adds at once all combinations of inputs and all permutations among each combination, such that each substage adds $\binom{n}{2}$ terms for two-way interactions and $\binom{n}{3}$ terms for three-way interactions. Then the sampler is called and the BIC or AIC is calculated. Because there is not a monotonic decrease / increase pattern for the objective function, a "tolerance" setting controls how many substages the algorithm can iterate through without finding a new minimum BIC or AIC before it terminates. The algorithm returns the optimum model.

This algorithm appears in the routine 'emulator_Xin' in the supplement.

## 3 Experiments: Dynamic system identification

### 3.1 Procedure

BSS-ANOVA regression – as is the case for other GPs – is most effective for tabular datasets with continuous inputs and targets of moderate dimensionality. This suggests an application in dynamic systems identification. Indeed BSS-ANOVA GPs have been utilized as components of other models ("intrusively") for this purpose in a number of applications [Bhat et al., 2017, Lei et al., 2019, Ostace et al., 2020]. We demonstrate here that they may also be used directly to identify dynamics in more general cases, without the aid of an accompanying model.

The procedure is a concurrent one, in that derivatives estimated from the datasets are modeled directly using BSS-ANOVA with forward variable selection, using the concurrent values of the system states and other inputs; for example a two-state system is modeled using two separate GPs:

$$\dot{x}_1 = \delta_1(x_1, x_2, u) \tag{28}$$
$$\dot{x}_2 = \delta_2(x_1, x_2, u) \tag{29}$$

The identified system is then integrated to yield predictions with uncertainty.

The procedure was demonstrated on two nonlinear dynamic datasets: a synthetic dataset derived from the susceptible, infected, recovered model (SIR model) for infectious disease, and the 'Cascaded

---
**Algorithm 1** BSS-ANOVA forward variable selection algorithm
---
1: **procedure** FWDVARSELECT($x$, $Z$, $\phi$, tol, $h$)  ▷ $h$ is a vector of hyperparameters.
2:   ind = 1
3:   count = 0
4:   **while** count < tol **do**
5:     **if** ind is new **then**
6:       Find all combinations of integers that sum up to ind, ordering them by the maximum
7:       integer appearing in each combination, with the lowest maximum first.
8:     1. Select the next combination in the set and place the integers into a vector with as many
9:     elements as there are model inputs, buffering out with zeroes.
10:     2. Produce a matrix $M_d$ the rows of which contain all permutations of that vector.
11:                               ▷ Each row corresponds to a term in the GP expansion.
12:     3. Produce an input matrix $X_d$ where columns are model terms and rows are experiments,
13:     for all terms appearing in $M_d$.
14:     4. Recursively concatenate: $X = [X X_d]$, $M = [M; M_d]$
15:     5. $\beta$, BIC = gibbs_Xin($X$, $Z$, $\phi$, $h$)
16:     **if** the BIC is a minimum for all models **then**
17:       save the model
18:       count = 0
19:     **else**
20:       count++
21:     **if** all combinations for ind have been utilized **then**
22:       ind++
23:   Return $M$, $\beta$, BIC
---

Tanks' experimental benchmark dataset. In both cases comparisons were made to long short term memory (LSTM) and gated recurrent unit (GRU) neural netowrks for timeseries prediction. In the case of the cascaded tanks benchmark comparisons were made against random forest (RF), a residual neural network (ResNet) and the state-of-the-art OAK inducing points scalable GP [Lu et al., 2022] for the static derivative estimation problem.

## 3.2 Experimental benchmark: Cascaded tanks

The cascaded tanks nonlinear benchmark dataset is an experimental nonlinear dynamic system [Wigren and Schoukens, 2013]. The experiment consists of a set of two tanks and a reservoir of water. An upper tank is filled by a pump from the reservoir. An outlet in the upper tank empties into the lower tank, which in turn empties through an outlet back into the reservoir. A signal sent to the pump serves as the forcing function for the system, with the tank water level heights the two states of the system.

We first compared the performance of BSS-ANOVA with RF, ResNet and OAK static regressors. Derivatives were calculated via direct finite differences for the relatively noise-free dataset, yielding 10000 instances. Each method was trained on concurrent values of both states and the forcing function for each derivative. For the GP we used hyperparameters of $a = 1000$, $b = 1.001$, $a_\tau = 4$ and $b_\tau = 55$ for $\dot{h}_1$ and 69.1 for $\dot{h}_2$, with tolerances of 3 for $\dot{h}_1$ and 5 for $\dot{h}_2$, and the AIC as discriminator. Of 2000 draws the first 1000 were discarded. Only two-way interactions were required. For the RF 100 trees were used with a leaf size of 5. The ResNet had a depth of 6 (filter sizes ranging from 16 to 64) and in between each fully connected layer is a batch normalization and relu layer. The mini batch size is 16, initial learn rate is 0.001, the data was shuffled every epoch for a total of 30 epochs, and the validation frequency was 1000. OAK was applied at a maximum dimension of 3 and with the default value of 200 inducing points. The 5-fold cross-validated results appear in Table 1. OAK performed best for both outputs, followed closely by BSS-ANOVA. Both GPs outperformed the RF and the ResNet by clear margins.

Timeseries predictions follow for the GP via a 4th-order Runge-Kutta integration routine. These were compared with LSTM and GRU recurrent neural networks (RNNs). For the LSTM there was one LSTM layer and a total of 128 hidden layers, the data was shuffled every epoch for a maximum of 125 epochs, verbose was equal to 0, and the sequence was padded to the left. The GRU had one

Table 1: Cascaded tanks 5-fold cross validated accuracies: derivatives

| Method | $\dot{h}_1$ (MAE/$10^{-4}$) | $\dot{h}_2$ (MAE/$10^{-4}$) |
|---|---|---|
| OAK | 17$\pm$4.7 | 36$\pm$2.4 |
| BSS-ANOVA | 18$\pm$6.5 | 39$\pm$3.6 |
| ResNet | 36$\pm$14 | 61$\pm$15 |
| RF | 30$\pm$9.4 | 49$\pm$4.9 |

Table 2: Cascaded tanks 5-fold cross validated accuracies: timeseries

| Method | $h_1$ (MAE/MAPE) | $h_2$ (MAE/MAPE) |
|---|---|---|
| BSS-ANOVA | 0.1167$\pm$0.0382 / 4.67$\pm$1.58 | 0.1577$\pm$0.0334 / 5.99$\pm$1.75 |
| LSTM | 0.2345$\pm$0.1006 / 9.46$\pm$4.87 | 0.2296$\pm$0.0378 / 9.58$\pm$3.32 |
| GRU | 0.3243$\pm$0.1092 / 12.16$\pm$5.02 | 0.2481$\pm$0.0402 / 9.89$\pm$3.40 |

GRU layer and 150 total hidden layers, the data was shuffled every epoch for a total of 150 epochs, verbose was equal to zero and the sequence was padded to the left. The 5-fold cross-validated results (datapoints were not randomized before creating the folds so as to preserve the timeseries order) appear in Table 2. BSS-ANOVA is most accurate, followed by the LSTM and the GRU. Figure 2 shows the predictions of the GP and the LSTM for the upper tank for one of the test folds. The GP predictions are superior near the sharp inflection and critical points where nonlinearities are strongest. Note that the first 50 points of each test set, which were provided to the LSTM and GRU as a start-up set in the prediction phase, were removed from the calculation of error for both methods.

While it is reasonable to expect that OAK with 200 inducing points would outperform BSS-ANOVA in the time integration, it was not practical to make this comparison for reasons of computing time. A comparison with a reduced number of inducing points and increased time step in the integrator was made – results are discussed in section 3.4.

### 3.3 Synthetic benchmark: Susceptible, infected, recovered model

The susceptible, infected, recovered model (SIR model) is a common simulation for infectious disease. Though there are several versions, the simplest is three states, only two of which are independent. The system is written

$$\dot{S} = -BIS/N_P \tag{30}$$

$$\dot{I} = BIS/N_P - \gamma I \tag{31}$$

$$\dot{R} = \gamma I \tag{32}$$

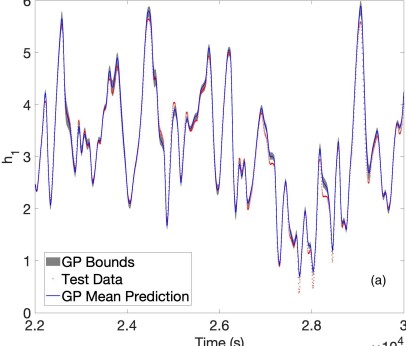
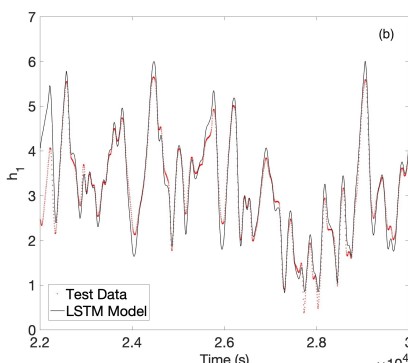

Figure 2: (a) BSS-ANOVA and (b) LSTM predictions vs. test set data for the water level height in tank 1 of the cascaded tanks dataset. Shaded regions in (a) are 95% confidence bounds as estimated from a draw of 40 curves.

where $S(t)$ is the susceptible population, $I(t)$ the infected, $R(t)$ the recovered, $B(t)$ is the transmissibility rate (which we utilize as a forcing function), $\gamma$ is the recovery rate (which we leave fixed at 0.5) and $N_P$ is the total population. Because $N_P$ is fixed and $S + I + R = N_P$, only two states are independent, so the system dynamics can be captured by modeling only two of the three. We chose $I(t)$ and $R(t)$.

The training data consists of 58 curves. All curves in the training set have a fixed $B$ value ranging from 0.5 to 9, in six intervals of 1.7. For each value of $B$ there are 8-10 siumulations corresponding to different initial conditions designed in such a way to provide coverage of the state space. (Exact initial conditions used appear in the supplement.) Each simulation used $N_P = 1000$.

The test data consists of 24 curves, each of which features a temporally changing transmissibility $B(t)$. There are three initial $B_0$ values: 1.35, 4.75 and 8.15. For each starting point there are two types of transmissibility curves: a ramp and a sinusoid. The $B_0 = 1.35$ and $B_0 = 4.75$ starting points have ramps with a positive slope of 1, while the $B_0 = 8.15$ curves have a slope of -1. All ramps run from t=0 to t=4, where they level off. The sinusoids have amplitudes between 0.5 and 3 and a period of 1.

Hyperparameters for BSS-ANOVA were: $a = a_\tau = 4$ for both states, $b_{\tau,R} = 8.95$ and $b_{\tau,I} = 72.1$, while $b_I = 1.25$ and $b_R = 20$. 2000 draws were taken and the first 1000 discarded. The tolerance was 6. Hyperparameters for the LSTM and GRU were the same as for the Cascaded Tanks.

A partial display of the results are shown in Figures 3 for BSS-ANOVA and 4 for the GRU, which was the better performing of the two neural nets on this dataset. For the GP, the total test set MAE was 5.2739±4.0138 for $I$ and 11.8345±21.7337 for $R$, corresponding to MAPEs of 8.99±4.92 for $I$ and 2.80±2.52 for $R$. Statistics were not calculated for the GRU as it failed to replicate the dynamics in most test cases and was obviously inferior in a quantitative sense in every instance, as shown in Figure 4.

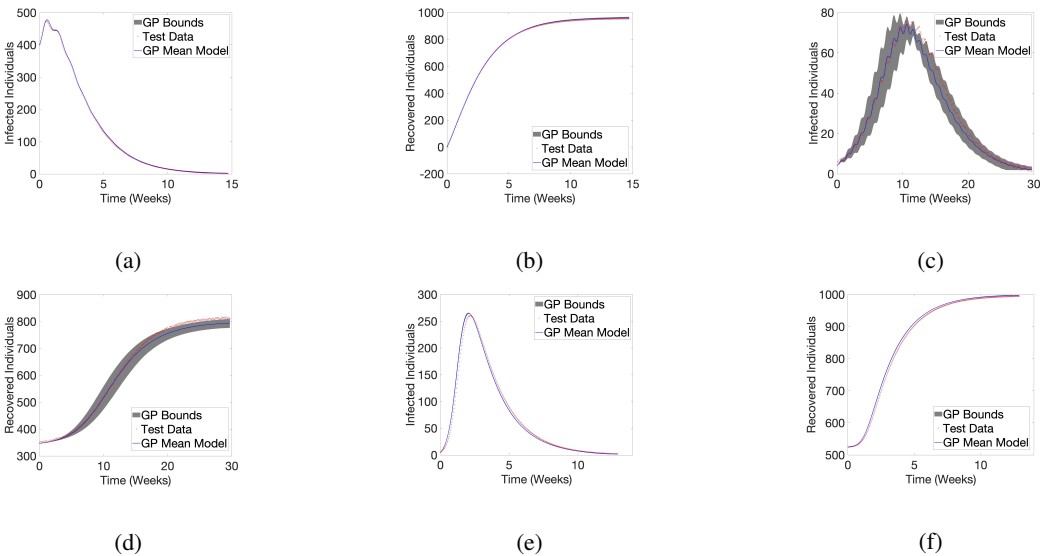

Figure 3: BSS-ANOVA results for 3 curves in the test set: (a)-(b) sine wave transmissibility with low initial infections; (c)-(d) sine wave transmissibility with moderate initial infections; (e)-(f) ramp transmissibility. Shaded regions are 95% confidence bounds for the predictions as estimated from a draw of 40 curves.

## 3.4 Training and inference times

Training and inference times for BSS-ANOVA were fast, with a mean total train time of 6.3 seconds for the cascaded tanks and 10.8 seconds for the SIR, with 8,000 and 20,000 training data points, respectively, on a 2019 6-core i7 processor with 16 GB of RAM. The routines were implemented in MATLAB, but not parallelized or optimized for speed. Models for $\dot{h}_1$ contain between 23 and 41

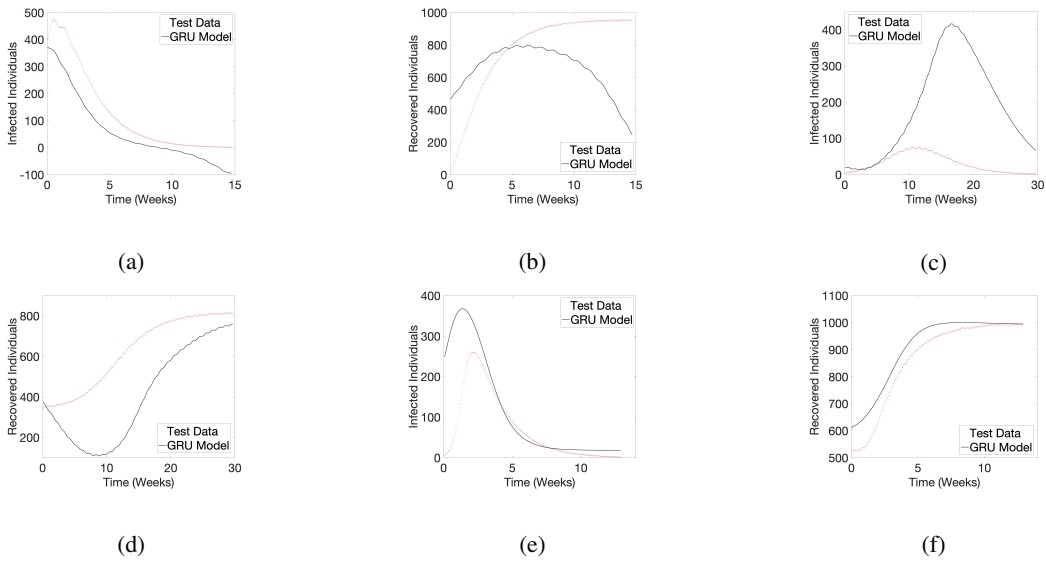

Figure 4: GRU results for 3 curves in the test set: (a)-(b) sine wave transmissibility with low initial infections; (c)-(d) sine wave transmissibility with moderate initial infections; (e)-(f) ramp transmissibility

terms, while $\dot{h}_2$ has between 38 and 57 terms. Prediction times for 2000 static points for the cascaded tanks averages 0.5437 s, and the time for evaluating integrals over the test set averages 20.22 s. For the SIR model the $\dot{I}$ model had 81 terms and the $\dot{R}$ model 9 terms, with a mean integration time of 5.3 s. Analyses have shown that the rate limiting step in BSS-ANOVA build algorithms are the $\mathcal{O}(NP)$ construction of the $X$ matrix from the inputs and basis functions. The neural networks were native MATLAB functions, parallelized and optimized for speed. Nonetheless train times were considerably longer, with mean train times of 130s for the ResNet and 175 and 123 s, respectively, for training the LSTM and GRU for the cascaded tanks. This is to be expected given that the number of weights in the neural nets are on the order of $10^4$.

It was not feasible to integrate OAK at the level of 200 inducing points to the same standard as that of BSS-ANOVA because of time considerations. A reduced set of 40 inducing points yielded accuracies in the static estimation problem that were approximately the same as BSS-ANOVA. A reduced time step (500 vs. 20,000 integration steps) brought the integration time down to 51 minutes for OAK, with MAE/MAPE of 0.1554/6.3 for $h_1$ and 0.2378/9.1 for $h_2$. Reducing the integration step to the same level as BSS-ANOVA (where we could expect comparable integration accuracies) would require approximately 33 hours.

## 4   Limitations and future work

The two examples presented in this paper were the only two attempted for dynamic systems identi-fication. Other benchmark dynamic systems, especially those chaotic in nature, will be examined in future work. Despite the advance in variable selection represented by this routine, datasets with higher dimensionalities in the feature space are more challenging and require additional methods for variable selection, which are already in development. More experiments and comparisons will be performed for dynamic systems as well, with larger datasets and more difficult identification problems. Like any GP BSS-ANOVA is inaccurate in extrapolation: when test set inputs exceed the bounds of the training set the resulting extrapolation sometimes causes instabilities causing the integration procedure to fail. These failures were eliminated by preventing extrapolation even in instances where the inputs exceeded the bounds, but more stable extrapolation strategies will possibly become necessary for longer prediction windows where extrapolation is unavoidable.

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
