# OpenReview forum: "Fast variable selection makes scalable Gaussian process BSS-ANOVA a speedy and accurate choice for tabular and time series regression"
_NeurIPS.cc/2022/Conference — NeurIPS 2022 Submitted_

### Official Review · Reviewer_j4ha · 2022-07-09

**Rating:** 4
**Confidence:** 3
**Soundness:** 3 good
**Presentation:** 1 poor
**Contribution:** 3 good

**Summary:**

The authors develop a Bayesian approach to  the Karhunen-Loéve (KL) decomposed kernel BSS-ANOVA

**Questions:**

See strengths and weaknesses

**Limitations:**

See strengths and weaknesses

**Strengths And Weaknesses:**

Strengths and Weaknesses:
Key weakness.  Most people in the GP community will not be familiar with the methods used.  So we need a _much_ clearer background, and perhaps more standard notation:
* Why are we using curly $\theta$ rather than $x$ for the input locations?
* Define kth Bernoulli polynomial.
* Tell us the shape of $\mathcal{B}_1$ etc.
* "the full kernel for a system with n inputs is written $\delta \sim MVN(0, \Gamma)$" is a very strange statement.  The kernel is written $\Gamma$, not $\delta \sim MVN(0, \Gamma)$?  Where did $\delta$ come from? Why are we using MVN rather than $\mathcal{N}$            (especially given that it can cause confusion with a matrix-variate Normal distribution.
* $\Gamma = \sigma_0^2 \tau_0^2 + ...$ makes no sense.  $\Gamma$ is a covariance matrix, while $\sigma_0^2 \tau_0^2$ are scalars?
* $\Gamma_{1, i}$ and $\Gamma_{2, ij}$ aren't defined.
* Where are the eigenvalues and eigenfunctions coming from?

Without these clarifications I do not believe most of the target audience will be able to understand the paper.

---

> ### Author Response · Authors · 2022-08-02
> **We will clear up notation and add comparisons to inducing point GPs**
>
> Thank you for the perceptive assessment of our work. We have tried to improve the presentation to make it more germane for the GP community in machine learning, and we have added comparisons to state-of-the-art inducing points-based scalable GPs.
>
> 1. $\vartheta$ is commonly used as notation for model inputs in our community but we are happy to switch to $x$.
> 2. We will present the formula for the $k^\textrm{\small th}$ Bernoulli polynomial, however instead of presenting them graphically we felt it was more appropriate to present the first few KL basis functions. This is the plan for the revision.
> 3. We will clean up the typos -- thanks for pointing them out.
> 4. The eigenvalues and eigenfunctions come from the eigendecomposition of $\Gamma_1$ as defined in (3), as evaluated for an input spanning the interval [0,1] and discretized on a dense grid with 500 intervals. This will be made clearer.

---

### Official Review · Reviewer_abxx · 2022-07-10

**Rating:** 5
**Confidence:** 3
**Soundness:** 3 good
**Presentation:** 3 good
**Contribution:** 2 fair

**Summary:**

The authors describe a method for variable selection in a GP type model for large datasets called BSS-ANOVA. This (existing) approach uses a kernel that has a basis decomposition where the basis functions can be precomputed and the training complexity becomes O(NP) where P is the number of terms from the kernel decomposition used. The papers’ contribution is on selecting the components in this decomposition with non-zero coefficients using the natural order of complexity and a forward selection paradigm. The coefficients and an information criterion is calculated via Gibbs sampling.

**Questions:**

What is the worst-case complexity of this algorithm? I.e. BSS-ANOVA has complexity O(NP) if I pre-specify the number of terms P, but your algorithm appears to determine that number dynamically. Would it naturally stop at N^2, could it be unbounded or is guaranteed to be better than N^2?

Is the Gibbs sampling algorithm described in Section 1.3 an original contribution of this paper or is it based on previous work? Does the crucial assumption (8) that enables the use of Gibbs sampling reduce the expressiveness of this model in a meaningful way or can it be shown that this assumption is wlog?

Please change the formula references to (N) to avoid confusion and clean up the citations ([Reich et al., 2009] vs Reich et al. [2009]).

Are there any other variable selection methods for BSS-ANOVA in the literature? Is there a way to use e.g. L1 penalties to find solutions faster?

**Limitations:**

I can not see any negative societal impact of this work. The authors do address some limitations in that the paper underperforms on some datasets.

**Strengths And Weaknesses:**

The paper describes a novel approach to fit a GP type model for large datasets and I think as such has a place in the literature.

I would like the authors to be more clear about the impacts of the proposed Gibbs sampler, as described below, i.e. both with regards to the novelty of it and whether it changes the fitted model. My assumption is that this method is new, but if it already exists in the literature, I do not think the use of an information criterion for variable selection alone is contribution enough to warrant acceptance.

---

> ### Author Response · Authors · 2022-08-02
> **We will clear up notation and add comparisons to inducing point GPs**
>
> Thank-you for this interesting perspective on our work. The method can indeed be thought of as a latent factor decomposition, where the mediating model is a Gaussian process. Similarly the combination of the Gibbs sampler and the variable selection routine can be thought of as an optimizer of a BIC/AIC objective function, which contain an $L^0$ penalty. We will emphasize this perspective in the rewrite, particularly the manner in which the forward variable selection algorithm manages dimensionality issues.
>
> 1. We have heard from other reviewers about notation and we are committed to making it clearer for the audience at NeurIPS. However we are confused by references to "objective criterion $J$" depending on $A$, $B$ and $D$, and a function "CalculateD." It is not clear to us which superscripts could be missing in (9).
>
> 2. The equality constraints in (8) are satisfied automatically in the Gibbs sampler through the use of a single prior for $\sigma$ and $\tau$. The same priors are used for each iteration in Algorithm 1.
>
> 3. We are not familiar with a method called FANC_OB, but we have added a new, high-performing inducing points GP (OAK) to the set of methodological comparisons.

---

### Official Review · Reviewer_jXjU · 2022-07-12

**Rating:** 4
**Confidence:** 3
**Soundness:** 3 good
**Presentation:** 3 good
**Contribution:** 2 fair

**Summary:**

Regarding the scalable Karhunen-Loeve decomposed kernel BSS-ANOVA for Gaussian process modeling, this paper proposed a BIC/AIC assisted forward variable selection method to have fast and accurate predictions for large tabular datasets. The performance of the proposed model was tested on two datasets.

**Questions:**

1) Additive Gaussian processes within the ANOVA formulation for high-dimensional inputs have been studied for a long history, see a recent paper
[1] Lu X, Boukouvalas A, Hensman J. Additive Gaussian Processes Revisited. arXiv preprint arXiv:2206.09861, 2022.
These however have not been discussed and compared in this paper.
2) Besides, the usage of BIC/AIC to perform truncation is not a new idea, what is the difference to previous works? Please highlight it.
3) What benefits can we have when using the ANOVA type decomposition for GP? Reducing computational budget? Figuring out import variables? The authors should make it clear rather than just showcasing the accuracy of time series regression.
4) The proposed model is suggested to be compared to the ANOVA-type GP competitors on more benchmarks to offer solid conclusions.


**Limitations:**

yes

**Strengths And Weaknesses:**

Pros:
1) A forward variable selection method via BIC/AIC is developed in the ANOVA regime for GP modeling;
2) The new method has been applied to time series regression.

Cons:
1) Benefits brought by forward variable selection are unclear;
2) Differences to existing works have not been well discussed;
3) Insufficient numerical experiments to showcase the superiority of proposed model.

---

> ### Author Response · Authors · 2022-08-02
> **We will add comparisons to the OAK inducing point GP**
>
> Thank-you for the attentive review of our paper. Here are our responses to your questions:
>
> 1. We have read the paper suggested and included comparisons with this method in our results. OAK outperformed BSS-ANOVA on the static estimation method for the cascaded tanks by slight margins (5-10%), using a small number of inducing points (30-40 for 2000 instances). However when we tried to incorporate OAK into the integrator it was not practical from a time standpoint: BSS-ANOVA is orders of magnitude faster.
>
> 2. Of course the use of BIC and AIC in model selection is not new, but this is not claimed as the novel aspect of the variable selection routine. The most important aspect is the exploitation of the ordering of basis functions to perform variable selection on the KL-decomposed kernel, which addresses the dimensionality issues facing KL-decomposed GPs. It is these decomposed terms which are selected, not the terms in the additive kernel. We will emphasize this in the revised text.
>
> 3. When combined with fast and automatic variable selection the new method outperforms many other methods in both speed and accuracy for low to moderate-dimensional, continuous tabular datasets. This combined with the speed of training and inference is why the application to dynamic systems identification is highlighted.
>
> 4. We compared with the OAK GPs on the dynamic datasets that appear in the original submission. We have also used BSS-ANOVA on other static datasets available in the UCI database, which the OAK authors reported results for. Our results look promising, however we chose not to report them in this work since our implementations for datasets with larger input dimensionalities use an additional variable selection methodology. The dynamic systems datasets are the kind of problems for which the method as presented is ideal: a small number (5 or fewer) of continuous inputs, coupled with an integrator that takes advantage of fast model evaluation. These advantages and limitations will be made clearer in the revised paper. We will introduce the additional variable selection methodology along with results on static tabular datasets in future contributions.

---

### Official Review · Reviewer_Yi9J · 2022-07-15

**Rating:** 3
**Confidence:** 4
**Soundness:** 1 poor
**Presentation:** 1 poor
**Contribution:** 1 poor

**Summary:**

The paper proposes a scalable GP based on Karhunen-Loeve expansions of kernels. The paper takes a fully Bayesian approach where Gibbs sampling is used to infer the kernel parameters, therefore, leading to forward variable selections.


**Questions:**

How to scale $\mu$ and $\Sigma$ in Eq. (10-11) which involve matrix inversion? If they are evaluated on 500 data points, such estimations are just local.


**Ethics Review Area:**

["I don’t know"]

**Limitations:**

Besides the presentation issues, I think the paper does not provide an extensive comparison with existing models. For example,an obvious baseline is sparse inducing GPs.

I believe the paper needs an additional revision to be ready for publication.


**Strengths And Weaknesses:**

Strengths:

- The variable selection using GP is an interesting problem.

I think the weaknesses outweigh the strengths, include

- The paper seems to be not well-structured, lacking a proper writing of what the problem that the paper tries to address here is.  It would be better to have an introduction section that would clarify those for us as readers.
- The paper does not provide sufficient descriptions or analyses for the method, for example, lacking literature reviews and no comparison with alternative methods.

---

### Author Response · Authors · 2022-08-02
**General comments on the reviews and proposed revisions**

We appreciate the efforts of the reviewers and the chairs in reading and commenting on our work. Most of us are new to the ML space and as such we are not sure about functional groupings -- it seems as if we ought to have emphasized the dynamic systems aspect of this work in the title, abstract and introduction as opposed to the 'tabular data' aspect. We believe the method is a promising one for tabular data, and indeed we have experimented with some of the UCI datasets with promising results. However success in that space will require additional variable selection methods for larger input space dimensionality and a mix of categorical and continuous inputs and targets. We have been working on such methods but have not presented them here -- as presented in the paper the method works best for continuous static datasets of small dimensionality (five or fewer).

We understand better than before that the community will be less familiar with Karhunen-Loéve decomposed GP kernels: while additive, ANOVA decomposed kernels may be common, KL decomposed kernels are not, as discussed in this recent preprint: arXiv:2108.05924v1. KL decomposed kernels have the potential to outperform inducing point scalable GPs in certain contexts since it avoids the reduction of the data inherent in selecting the inducing points. They are also highly interpretable, since the contribution of each input and input combination is clear from the model form. However the advantage we would like to highlight in this contribution is the manner in which forward variable selection greatly improves the speed of training and inference of KL decomposed GPs, such that the method becomes highly adaptive to intrusive applications in domain science, particularly dynamic systems.

As discussed in the above-referenced paper, the main challenges of KL decomposed kernels are the estimation of the basis functions (which can be computationally onerous) and dimensionality issues arising from the KL expansion. The former issue has been dealt with already for this kernel by Brian Reich and co-workers, who used a dense grid for discrete basis function evaluation, which we fitted to splines. The main contribution of this work is dealing with the dimensionality issue, which we handle by taking advantage of the strongly ordered nature of the basis functions in a forward variable selection routine incorporating $L^0$ penalties (inherent in the BIC/AIC functions). The resulting models are fast, both in training and inference -- orders of magnitude faster than the fastest inducing point GPs. This strongly suggests applications in dynamic systems, as the GP can be trained on state derivatives and then used in an integrator, and more generally in an intrusive fashion in domain-specific modeling contexts.

We propose to revise the paper to emphasize these aspects in title, abstract and introduction, along with other changes outlined in responses to individual reviewers.

---

### Meta-Review · Area_Chair_4iLn · 2022-08-25

**Recommendation:** Reject
**Confidence:** Certain

**Metareview:**

Reading the reviews, I think there are ultimately two challenges for the authors to address in this work. First I think ends up being a somewhat simple "background for the community" problem, as both several reviewers and the authors in their general comments point out: significantly more background on KL decomposed kernels may be warranted, and this lack of background (along with perhaps some simple notational differences) led to what I felt like were some challenges understanding the full paper.

With that being said, I don't think the above alone should be sufficient by itself to result in rejection, despite this lack of context likely being a primary contributor to final review scores. However, I do agree with reviewers' concerns that there are some concrete comparisons to existing scalable GP literature missing. I think the inclusion of OAK + inducing points is a start, but e.g. the use of m=40 inducing points in section 3.4 is surprising to me, partly perhaps because it's not clear which task this was a problem for (you state early that you use m=200 for the cascaded tanks task), and partly because none of the dataset sizes involved appear to me to be even remotely beyond the capability of these existing scalable GP approximations in the literature -- even up to m=512 or m=1024 inducing points is fairly standard practice. Given the reasonably good performance of OAK in some of the new results even with limited inducing point sets, I think that clearly further investigation is warranted there. Beyond inducing point methods, there are also NNGP / Vecchia models (which have also been recently made variational via Wu et al., 2022), and even exact GPs seem readily applicable to some of the tasks considered here with access to even a single moderately powerful GPU.

**Award:**

No

---

### Decision · Program_Chairs · 2022-09-14

Reject